

# Changes in the concentration of bone turnover markers in men after maximum intensity exercise

Małgorzata Bagińska[1], Łukasz Marcin Tota[2], Małgorzata Morawska-Tota[3], Justyna Kusmierczyk[2] and Tomasz Pałka[2]

[1] Faculty of Physical Education and Sport, University of Physical Education in Kraków, Kraków, Poland
[2] Department of Physiology and Biochemistry, University of Physical Education in Kraków, Kraków, Poland
[3] Department of Sports Medicine and Human Nutrition, University of Physical Education in Kraków, Kraków, Poland

Corresponding author
Małgorzata Morawska-Tota, malgo-rzata.morawska@awf.krakow.pl

## ABSTRACT

**Background.** Physical activity is an important factor in modelling the remodelling and metabolism of bone tissue. The aim of the study was to evaluate the changes in indices demonstrating bone turnover in men under the influence of maximum-intensity exercise.

**Methods.** The study involved 33 men aged 20–25, divided into two groups: experimental ($n = 15$) and control ($n = 18$). People training medium- and long-distance running were assigned to the experimental group, and non-training individuals to the control. Selected somatic, physiological and biochemical indices were measured. The level of aerobic fitness was determined using a progressively increasing graded test (treadmill test for subjective fatigue). Blood samples for determinations were taken before the test and 60 minutes after its completion. The concentration of selected bone turnover markers was assessed: bone fraction of alkaline phosphatase (b-ALP), osteoclacin (OC), N-terminal cross-linked telopeptide of the alpha chain of type I collagen (NTx1), N-terminal propeptide of type I progolagen (PINP), osteoprotegerin (OPG). In addition, the concentration of 25(OH)D3 prior to the stress test was determined. Additionally, pre and post exercise, the concentration of lactates in the capillary blood was determined.

**Results.** When comparing the two groups, significant statistical differences were found for the mean level of: 25(OH)D3 ($p = 0.025$), b-ALP ($p < 0.001$), OC ($p = 0.004$) and PINP ($p = 0.029$) prior to the test. On the other hand, within individual groups, between the values pre and post the stress test, there were statistically significant differences for the average level of: b-ALP ($p < 0.001$), NTx1 ($p < 0.001$), OPG ($p = 0.001$) and PINP ($p = 0.002$).

**Conclusion.** A single-session maximum physical effort can become an effective tool to initiate positive changes in bone turnover markers.

## INTRODUCTION

Bone tissue undergoes constant changes, remaining in a state of dynamic balance between bone-forming and bone-destroying processes. These processes occur throughout life and are called bone turnover. When resorption processes predominate over bone formation, homeostasis is disturbed and negative balance is created, manifested by loss of bone mass, leading to increased bone fragility (*Sipos, Pietschmann & Rauner, 2008*; *Langdahl, Ferrari & Dempster, 2016*; *Compston, McClung & Leslie, 2019*). The process of bone remodelling is controlled by many factors, including gender, somatic indices (body mass, ratio of muscle mass to lean body mass, body height, BMI –body mass index), diet (*e.g.* protein and calcium supply), medications, stimulants, diseases or physical activity (*Sinaki, 2010*; *Andreoli et al., 2012*; *Kujawska et al., 2019*).

Changes in the bone tissue may be typical and characteristic of the selected type of physical effort. Regular and properly selected physical activity causes the activation of bone adaptation in places where the load is applied. However, not every type of training load will stimulate remodelling and increase bone parameters to the same extent (*Bednarski et al., 2018*). The bone is modelled under the influence of skeletal muscles and gravity (*Leigey et al., 2009*; *Willigenburg, Kingma & Van Dieën, 2013*). The positive impact of mechanical loads has been confirmed in numerous scientific studies (*Hojan & Milecki, 2012*; *Andreoli et al., 2012*; *Krahenbühl et al., 2014*; *Duda & Wójtowicz, 2014*; *Weaver et al., 2016*). There are also studies in which it is shown that practicing competitive sport has beneficial effects on bone tissue, both during one's sports career and after its completion (*Andreoli et al., 2012*; *Kaźmierczak et al., 2015*). However, athletes, due to the mere fact of increased physical activity generated by training, are not protected against the occurrence of disorders in the metabolism of bone tissue. The reason for this is too high intensity and volume of training units stretched over the course of an athlete's career (*Oosthuyse, Badenhorst & Avidon, 2014*; *Mcveigh et al., 2015*; *Mojock et al., 2016*). Based on numerous studies, it has been demonstrated that an insufficient level of physical activity, sentential lifestyle and immobilisation lead to significant loss of bone mass (*Ćwirlej & Wilmowska-Pietruszyńska, 2008*; *Nawrot-Szołtysik, Zmudzka Wilczek & Doroniewicz, 2010*; *Saraví & Sayegh, 2013*; *MacKnight, 2017*). In population studies, it has been indicated that physically active people have higher bone mass and better mineralised bone tissue compared to inactive people (*Dionyssiotis et al., 2010*; *Von Stengel et al., 2011*; *Karlsson & Rosengren, 2012*; *Camhi & Katzmarzyk, 2012*).

Maximum exercise intensity is a key aspect of improving physical performance and achieving peak performance in various sports (*Gibala et al., 2012*). Analysing the impact of maximum intensity exercise on the human body has been particularly applied within the context of running (*Ouerghi et al., 2017*). The great significance of high-intensity interval training (HIIT) in the context of bone turnover is confirmed by research, but the role of selected markers remains unclear and requires further enquiry (*Kouvelioti et al., 2018*).

To optimise the impact of physical activity on bone health, it is necessary to understand the response of bone tissue to specific mechanical stimuli (*Maimoun, 2005*). Knowledge regarding the mechanisms regulating bone mineralisation can contribute to the prevention

and reduction concerning the risk of musculoskeletal injuries in physically active individuals (*McArdle, Katch & Katch, 2010*).

In earlier research conducted by *Nowak et al. (2020)*, the authors used training intensities in the 90% VT range for 60 minutes, three times a week and for a period of 12 weeks. In turn, the research conducted by *Guerriere et al. (2018)* was based on resistance training using a dynamometric platform, where the intensity was individually selected at the level of 40% of 1-RM. The participants performed 10 sets of 10 repetitions, once a week, for 4-6 weeks. However, the research conducted by *Sherk et al. (2017)* was focused on training with a set intensity equivalent to 60–75% of maximal oxygen consumption over a distance of 35 km using a bicycle ergometer.

The measurement of biochemical marker concentrations for bone tissue remodelling allows to illustrate the osteogenic and osteolytic processes taking place in bone tissue. Simultaneous determination of bone marker synthesis and resorption further allows to visualise the intensity and pace of opposing processes. Thus, increasing the identification of people at an increased risk of fractures (*Riggs, 2000*; *Delmas et al., 2000*; *Baczyk, Chuchracki & Klejewski, 2012*; *Drwęska-Matelska et al., 2014*).

Rational nutrition is a significant factor affecting bone tissue composition. In order to maintain homeostasis of the body, a properly balanced diet rich in alkaline-forming products should be used. At the same time, acidifying products should be limited (*Dardzińska, Chabaj-Kędroń & Małgorzewicz, 2016*; *Kwiatkowska, Lubawy & Formanowicz, 2019*). The most important components of such a diet are considered proteins, vitamin D and calcium. When choosing products that are a source of protein, the ratio of protein to fat content should be taken into account. This is because it affects acid-base balance (*Fraczek, Krzywański & Krzysztofiak, 2019*). Additionally, adequate vitamin (A, C, K) and mineral (Cu, Zn, Fe, Mg, P) supply has positive effects on the metabolism of bone tissue and its mineralisation (*Platta, 2014*). Physically active people are exposed to low energy availability (EA) due to an insufficient supply of nutrients with diet and exercise energy expenditure (*Loucks, Kiens & Wright, 2011*). There is a risk that among athletes practicing endurance sports, there may be an insufficient supply of energy, which may have adverse effects on the bones (*Loucks, Kiens & Wright, 2011*; *Slater et al., 2016*; *Papageorgiou et al., 2018*). Low energy availability is directly related to low bone mass, disturbed bone microarchitecture, and thus, an increased risk of stress-related fractures (*Barrack et al., 2014*; *Southmayd et al., 2017*).

Maximum effort levels go beyond typical training protocols, which can lead to faster and more noticeable responses in bone metabolism. In this way, this study enriches our knowledge about extreme loads and their impact on bone remodelling processes.

The aim of the study was to assess changes in: b-ALP, OC, NTx1, PINP and OPG as selected markers illustrating dynamic phenomena in bone tissue transformation (bone turnover) among men subjected to maximal intensity exercise.

The potential implications of this study extend to both the sports and health care domains, offering specific guidance for clinical and training practice.

## MATERIAL AND METHODS

### Characteristics of the study group

The study covered a group of 33 healthy men aged 20–25. The subjects were divided into two groups. The experimental group ($n = 15$) comprised individuals training medium- and long-distance track-and-field runs, while the control group ($n = 18$) did not engage in regular physical activity. The term "non-training people" referred to individuals not undertaking regular physical activity, such as organised sports training or other forms of individual physical activity. These people may have led a sedentary lifestyle or did not engage in physical activity due to various reasons, such as lack of motivation, health limitations or lack of time. The inclusion and exclusion criteria for/from the study are presented in Table 1. Using a test power of 80%, a confidence level of 95% and an effect size of 0.4, the sample size was determined to be 15 participants in each group. Analyses were conducted using G*Power 3.1 software (Düsseldorf, Germany). The statistical analyses were performed using the IBM SPSS Statistics version 28 software package (IBM Inc., Armonk, NY, USA).

The study project was approved by the Bioethics Committee at the District Medical Chamber in Kraków, No. 319/KBL/OIL/2021. The subjects were informed about the aims as well as course of the research and provided their written consent for participation in the research.

### Research plan

The study organisation plan included: qualitative and quantitative assessment of the subjects' diet, anthropometric measurements, assessment of aerobic capacity and a biochemical blood test (Fig. 1). All physiological and biochemical tests were performed in an air-conditioned laboratory in the morning, no earlier than 2 hours after consuming a light meal.

### Anthropometric measurements

Prior to the beginning the physiological and biochemical tests, selected somatic indices were measured. The level of indicators allowing for the analysis of body composition was determined using the AKERN BIA 101 body composition analyser. Body mass (BM) was measured using the SECA 875 Portable Medical Scale to the nearest 0.1 kg. The bioelectrical impedance technique was applied to assess body structure by determining fat free mass (FFM), fat mass (FM) and percentage of body fat (FM%). Body height (BH) was determined using the Seca 213 stadiometer, with a measurement accuracy to the nearest one mm.

### Graded test

To assess aerobic capacity, a direct method was used—a test with a gradually increasing load performed on a mechanical treadmill (Saturn 250/100R, h/p/Cosmos, Germany) until refusal to continue work due to extreme fatigue. The test allowed to determine physiological indicators at the level of the second ventilatory threshold (VT2) and at the maximum level (VO$_2$peak). Data were collected as previously described in *Tota et al. (2021)*. Specifically

**Table 1  Inclusion and exclusion criteria.**

| Experimental group | Control group | Inclusion criteria | Exclusion criteria |
|---|---|---|---|
| x | x | Male gender | Smoking |
| x | x | Age between 20 and 25 years | Abusing alcohol or other stimulants |
| x | x | Medical certificate confirming the lack of health contraindications to perform exercise | |
| x | | Endurance nature of physical of physical activity: medium- and long-distance runs | |
| x | | Min. 5 years of training experience | |
| x | | At least participation in national and Polish, nationwide competitions | |
| x | x | Not using pharmacotherapy or supplements during the study period and at least 4 weeks prior to the intervention | |
| | x | Lack of undertaking regular physical activity | |
| x | x | Conscious consent of the patient to take part in the study | |

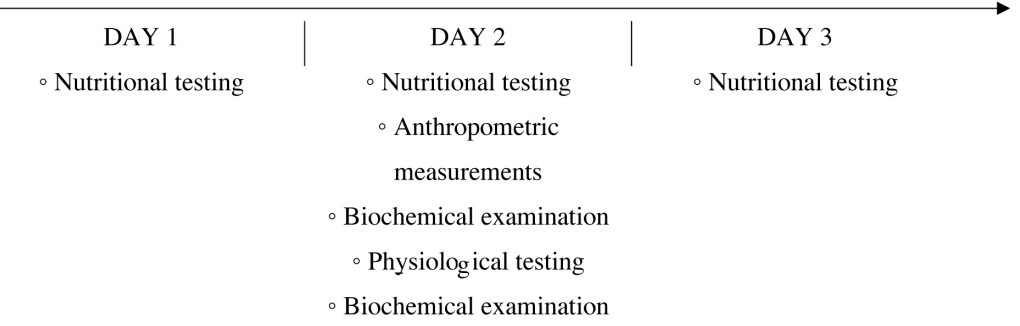

**Figure 1  Scheme of the research organisation.**

in order to determine VT2, changes in respiratory indices with increasing work intensity were analysed. The criteria for determining VT2 were as follows: (a) the percentage of $CO_2$ in the exhaled air reached the maximum value and then decreased, (b) the respiratory equivalent for carbon dioxide reached the minimum value and then increased, (c) after exceeding VT2, a large non-linear increase in pulmonary ventilation was noted (*Tota et al., 2021*). The $VO_2$ peak value was considered the highest of the recorded values (*Binder et al., 2008*).

The test effort began with a 4-minute warm-up during which the subject ran at a constant speed of 8 km h$^{-1}$, with a treadmill inclination angle of 1°. Then, the running speed was increased by 1.0 km h$^{-1}$ every 2 minutes. The test was performed until refusal to continue further work due to extreme fatigue.

During the test, the following indices were recorded using an ergospirometer (Cortez Metalyzer 3B, Germany): lung ventilation per minute, percentage of carbon dioxide in exhaled air, oxygen uptake per minute, carbon dioxide excretion per minute, respiratory quotient and respiratory equivalent for carbon dioxide. Heart rate (HR) during the test was measured using a sport-tester (H7, Polar, Finland).
During the entire duration of the graded test and 1 hour after its completion, the subjects did not consume any meals or fluids.

All tests were performed at the Central Scientific and Research Laboratory (CLNB) of the University of Physical Education in Kraków, PN-EN ISO 9001:2015.

## Biochemical measurements

Blood for biochemical determinations was collected from the elbow cavity by a qualified laboratory diagnostician 1 hour pre-test and 60 minutes post its completion, in conditions of a certified laboratory (PN-EN ISO 9001:2015) and in accordance with applicable standards. Blood was collected into Vacutainer EDTA tubes. The blood samples were centrifuged (2,000 rpm, 15 minutes) (MPW-351R centrifuge, Poland) to separate the serum.

The concentration of biochemical markers in the serum was determined *via* Enzyme-linked immunosorbent assay (ELISA) with absorbance measurement using the Spark® multimode microplate reader (Tecan, Grödig, Austria). The ELISA procedure was performed in duplicate to enhance replicability. The results were calculated according to manufacturer manuals (ELK Biotechnology, Wuhan, China). Intra-assay Precision (precision within an assessment): CV% <8%. Three samples of known concentration were tested 20 times on 1 plate to assess intra-assay precision. Inter-assay Precision (precision between assays): CV%<10%. Three samples of known concentration were tested in 40 separate assays to assess inter-assay precision.

Reagent kits (ELK Biotechnology, Wuhan, China) were used to determine markers of bone turnover: bone fraction alkaline phosphatase (ELK8560), osteoclacin (ELK2390), N-terminal cross-linked telopeptide of the alpha chain of type I collagen (ELK1843), N-terminal propeptide of type I procollagen (ELK5402), osteoprotegerin (ELK1176). The concentration of the metabolite 25(OH)D3 (K2108-190708) was determined only once (before the exercise test) using the Immundiagnostik AG (Stubenwald, Germany) reagent kit.

Blood for the measurement of lactate concentration was collected from the fingertip pre-beginning the graded test and at 3 and 20 minutes post its completion. Lactate concentration was determined *via* the Dr. Lange LP20 (Germany) (520 nm measuring wavelength).

## Methodology of nutritional testing

In order to carry out qualitative and quantitative assessment of the diet, the subjects kept food diaries for 3 days, on the day before the physiological and biochemical examination, on the day of the examination and the day after. The supply of energy, proteins, fats, total carbohydrates, as well as selected vitamins and minerals, was estimated in relation to the current standards of the National Institute of Public Health (PZH –PIB) (*Jarosz et al., 2020*). Qualitative and quantitative assessment of the consumed nutrients was carried out on the basis of the ALIANT dietary computer program (Poland, 2024, Design by EXPROMO).

## Statistical analysis and manner of presenting the results

Statistical analysis was performed in R ver. 4.1.3. The Shapiro–Wilk test was used to assess the normality of continuous variable distribution. The Student's $t$-test or its non-parametric equivalent (the Mann–Whitney U test) was used to compare the two groups. The effect size was calculated, where in the case of parametric test comparisons, the applied effect size measure was the value of Cohen's d. When non-parametric tests were used, the $\eta^2$ value was calculated. In the interpretation of the d value regarding effect size, the following values were adopted: 0.2—weak, 0.5—moderate, above 0.8—strong. For interpretation of the $\eta^2$ value regarding the effect size, the following were assumed: 0.01—weak, 0.06—moderate, over 0.14—strong. The test probability of $p<0.05$ was considered statistically significant.

## RESULTS

In Table 2, the characteristics are presented regarding the somatic indices of the studied groups. Significant statistical differences were observed in all analysed somatic indicators, except for body height. The test probability of $p < 0.05$ was considered significant (Table 2). As a result of the conducted test, a significantly higher ($p = 0.017$) value of lung ventilation per minute was recorded among the training participants. Detailed changes in physiological indices under the influence of the graded test are presented in Table 3.

Changes in bone turnover markers under the influence of a single exercise session at maximum intensity are presented in Fig. 2. Significant statistical differences were found for the average level of: $25(OH)D_3$ (*p = 0.025*), b-ALP ($p < 0.001$), OC (*p = 0.004*) and PINP ($p = 0.029$) pre the stress test. In turn, following the test, significant differences were noted for the average level of: b-ALP ($p < 0.001$), NTx1 ($p < 0.001$), OPG ($p = 0.001$) and PINP ($p = 0.002$) among the examined groups of men (Fig. 2).

The study groups differed statistically significantly w regard to: b-ALP (*p = 0.025*), OC (*p = 0.031*) and NTx1 (*p = 0.034*) (Table 4).

No effect of diet was noted on the changes in markers of bone turnover.

## DISCUSSION

To our knowledge, this study is the first in which the effects of a single-session maximum intensity exercise have been evaluated with regard to changes in the concentration of bone turnover markers among training and non-training subjects. In the literature on the subject, the only works that can be found are on changes in bone turnover markers under the influence of physical efforts of different nature and duration (*Sherk et al., 2017*; *Sansoni et al., 2018*; *Guerriere et al., 2018*; *O'Leary et al., 2019*; *Nowak et al., 2020*).

In recent years, there has been significant progress research on the biochemical indices of bone turnover, both in osteogenic and osteolytic processes. For the diagnosis and prediction of complications in the area of bone tissue, *e.g.* biochemical markers of bone turnover (BTM) are used (*For the IOF-IFCC Bone Marker Standards Working Group et al., 2011*; *Garnero, 2014*; *Mielnik, Świetochowska & Ostrowska, 2019*).

Applying mechanical stress (physical activity) to the skeleton can positively affect bone mineral density (BMD). In the literature on the subject, the positive effect of long-term

**Table 2  Characteristics of somatic indices among the studied men.**

| Index | Training individuals | Non-training individuals | *p* | Effect size |
|---|---|---|---|---|
| BH [cm] | 182.01 ± 3.53 | 179.42 ± 6.83 | *0.175* | *d* = 0.460 |
| BM [kg] | 75.84 ± 7.66 | 77.11 ± 8.10 | 0.864 | $\eta^2$ = 0.001 |
| FFM [kg] | 62.66 ± 4.06 | 67.23 ± 6.91 | **0.036** | *d* = 0.781 |
| FM [%] | 17.04 ± 4.27 | 12.73 ± 3.70 | **0.005** | *d* = 1.092 |
| FM [kg] | 13.17 ± 4.52 | 9.87 ± 3.19 | **0.022** | *d* = 0.863 |

Notes.

BH, body height; BM, body mass; FFM, fat free mass; FM, fat mass; F%, percentage of body fat.
Significance level $p < 0.05$.
Italics indicate statistical significance. Statistically significant values are in bold.

**Table 3  Level of selected indices characterising aerobic capacity.**

| Index | Training individuals | Non-training individuals | *p* | Effect size |
|---|---|---|---|---|
| Maximum values | | | | |
| MAX t [min] | 17.67 ± 2.88 | 16.60 ± 2.74 | *0.292* | *d* = 0.366 |
| v [km/h] | 16.06 ± 2.25 | 14.94 ± 1.63 | *0.115* | *d* = 0.570 |
| HR [b/min] | 184.86 ± 7.92 | 188.61 ± 13.76 | *0.341* | *d* = 0.206 |
| $VO_2$ [L·min$^{-1}$] | 3.92 ± 0.24 | 3.99 ± 0.64 | 0.672 | *d* = 0.139 |
| $VO_2$/kg | 52.04 ± 4.99 | 52.14 ± 9.00 | 0.732 | $\eta^2$ = 0.004 |
| VE [l/min] | 156.64 ± 18.65 | 150.03 ± 23.20 | *0.392* | *d* = 0.289 |
| Values for VT2 | | | | |
| VT2 t [min] | 10.84 ± 1.75 | 10.50 ± 1.99 | 0.619 | *d* = 0.181 |
| HR [b/min] | 166.21 ± 6.39 | 169.78 ± 10.10 | *0.119* | $\eta^2$ = 0.029 |
| $VO_2$ [L·min$^{-1}$] | 3.31 ± 0.34 | 3.30 ± 0.61 | *0.382* | $\eta^2$ = 0.055 |
| $VO_2$/kg | 43.91 ± 4.94 | 44.30 ± 7.67 | 0.871 | *d* = 0.210 |
| VE [l/min] | 102.85 ± 16.83 | 89.82 ± 12.42 | ***0.017*** | *d* = 0.940 |
| II vent. thresh. %$VO_2$max | 84.49 ± 6.36 | 82.46 ± 6.59 | *0.387* | *d* = 0.506 |
| II vent. thresh. %HR | 89.98 ± 3.21 | 90.37 ± 7.69 | *0.393* | $\eta^2$ = 0.046 |
| Lactate concentration in the blood | | | | |
| La 0 | 1.52 ± 0.19 | 1.54 ± 0.15 | *0.819* | $\eta^2$ = 0.005 |
| La 3 | 11.92 ± 2.50 | 13.13 ± 2.51 | *0.184* | *d* = 0.383 |
| La 20 | 6.06 ± 2.15 | 6.89 ± 2.47 | *0.635* | $\eta^2$ < 0.001 |

Notes.

MAX t, maximal running time; v, running speed; HR, heart rate; $VO_2$, global oxygen uptake (L·min$^{-1}$); $VO_2$/kg, oxygen uptake relative to body mass (mL min-1 kg$^{-1}$); VE, ventilation per minute; VT2, second ventilatory threshold; La, blood lactate concentration.
0 - baseline, 3' - in the 3rd minute after the end of the graded test, 20' - in the 20th minute after the end of the test.
Italics indicate statistical significance. Statistically significant values are in bold.

physical training on BMD in the elderly is well-described, it has been shown that it depends on the intensity and nature of the physical effort (*Bolam, Van Uffelen & Taaffe, 2013*; *Daly et al., 2019*). Therefore, the impact of vigorous, short-term physical exercise on changes in markers of bone turnover is unclear, as the existing results regarding the impact of various mechanical stimuli on BTM are conflicting (*Delmas et al., 2000*; *Smith et al., 2021*).
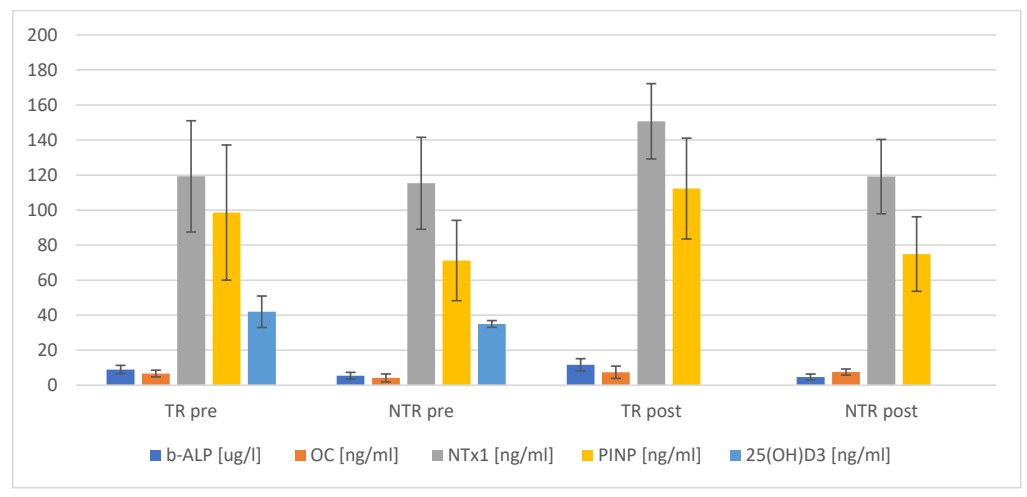

**Figure 2 Changes in selected markers of bone turnover, comparison of means between 2 groups.** TR, training; NTR, non-training.

**Table 4 Value of changes in bone turnover markers pre- and post-exercise Δ.**

| Markers | Training individuals | Non-training individuals | *p* | Effect size |
|---|---|---|---|---|
| b-ALP [ug/l] | −0.794 ± 2.55 | 2.724 ± 5.57 | ***0.025*** | $d = 0.779$ |
| OC [ng/ml] | −0.721 ± 2.19 | 1.412 ± 2.95 | ***0.031*** | $d = 0.805$ |
| NTx1 [ng/ml] | −16.440 ± 17.78 | −3.828 ± 14.29 | ***0.034*** | $d = 0.793$ |
| OPG [ng/ml] | −0.192 ± 5.34 | −0.352 ± 0.403 | *0.970* | $\eta^2 < 0.001$ |
| PINP [ng/ml] | −1.537 ± 5.17 | −2.396 ± 6.93 | *0.702* | $d = 0.138$ |

**Notes.**

b-ALP, bone alkaline phosphatase; OC, osteocalcin; NTx1, N-terminal telopeptide of collagen type I; OPG, osteoprotegerin; PINP, procollagen I N - terminal propeptide; Δ, differences in changes between the mean value from before and after the stress test.

Italics indicate statistical significance. Statistically significant values are in bold.

Based on the results of the authors' research, an increase in the concentration of all osteogenic markers (b-ALP, OC, PINP) in the group of athletes was shown, on average, by 29.4%, 10.8% and 13.9%. In the group of non-training individuals, there was no increase in the bone fraction of alkaline phosphatase. This observation could potentially indicate an immediate anabolic effect of maximum intensity exercise on bone tissue (*Mezil et al., 2015*; *Falk et al., 2016*). *Maïmoun et al. (2006)*, examining the impact of physical exercise intensity level on bone metabolism, indicated a periodic increase in OC and b-ALP for both levels of the set intensity, which could suggest the existence of a bone tissue stimulation threshold. *Nishiyama et al. (1988)* found that a 30-minute moderate-intensity run resulted in a significant increase in blood OC concentrations that varied in response time depending on the level of training. A similar tendency could be observed in the authors' research, where the initial OC concentration was higher in the training athletes (6.68 ng/ml *vs.* 4.14 ng/ml), and post the stress test, a higher value was observed in the group of non-training individuals (7.53 ng/ml *vs.* 7.40 ng/ml). However, in the graded test, a longer running time was observed

among athletes, on average, by 1.07 min, which could suggest that the duration of exercise is also an important indicator in the response of bone markers to acute exercise (*Wallace et al., 2000*). The results of research regarding the effects of exercise on bone alkaline phosphatase are inconsistent. In the literature, there are both reports of a direct increase in b-ALP immediately, 30 minutes or 24 hours post exercise (*Rudberg et al., 2000*; *Maïmoun et al., 2006*; *Kish et al., 2015*), and those with no significant changes noted (*Guillemant et al., 2004*; *Scott et al., 2010*). Such a position corresponds to the results of the authors' research, in which a 14.4% decrease in the concentration of b-ALP among non-training people and a 29.4% increase in athletes was observed immediately following the test. Brahm et al. theorised that changes in the concentration of bone markers determined in serum post vigorous exercise could be explained by the process of haemoconcentration (*Brahm, Piehl-Aulin & Ljunghal, 1997*). This hypothesis was not confirmed in the most recent research in which it was stated that these changes, as a response to physical exercise, are higher than those directly resulting from haemoconcentration. Considering the relatively short duration of the experiment, one might consider whether *de novo* synthesis took place (*Wallace et al., 2000*).

The level of physical activity is another factor that may influence variation in the level of bone alkaline phosphatase (b-ALP) activity in response to physical exercise (*Liu et al., 2022*), which could also be observed in our study. However, in research conducted by *Colaianni et al. (2014)*, it was proved that after 3 weeks of induced exercise, there is an increased differentiation of osteoblasts *in vitro* compared to resting conditions, which would indicate a positive effect of irisin visible during physical exercise by working muscles on bone tissue (*Colaianni et al., 2014*). *Zhang et al. (2017)* indicated that 2 weeks of exercise (running) increases the expression of osteogenic markers in bone tissue (*Zhang et al., 2017*). Prescribed exercise improves bone microarchitecture and increases the number of ALP-positive cells in mice (*Zhao et al., 2021*).

Bone resorption markers may be elevated in long-distance runners engaging in daily activity and in athletes who have suffered stress fractures in the past, even if there is no current risk of such fractures (*Fujita et al., 2017*). In our research, significant differences were demonstrated in the mean level of osteolytic marker (NTx1 $p < 0.001$) post a stress exercise. Moreover, there was a statistically significant change in the NTx1 level pre and post the test in the training group, which was not observed in the control group. The discipline practiced (medium- and long-distance runs) and the sports level determining the number and volume of training units could have resulted in an increase in the value of the oseolytic marker. Unfortunately, the authors of this study did not verify whether there were any cases of overuse injuries in the past among the examined athletes and whether this could have resulted in such a large increase in the analysed indicator. One of the important factors affecting the proper structure and functioning of the skeletal system is $25(OH)D_3$, which differentiates osteoblasts and proliferates pre-osteoblasts (*Fretz et al., 2007*). It also regulates calcium metabolism in the body. In the authors' study, a significantly higher ($p = 0.025$) average concentration of $25(OH)D_3$ was observed among the tested athletes. The concentration of calcidiol in both study groups was within the normative values (*EFSA Panel on Dietetic Products, Nutrition and Allergies, 2012*). Further in the authors' research,
no positive correlation was found between the level of 25(OH)D$_3$ and the concentration of OC and OPG. Shimasaki et al., conducting research among footballers, observed that too low levels of 25(OH)D$_3$ in blood serum (<30 ng/ml) induced a significant increase in the risk of 5th metatarsal fracture (*Shimasaki et al., 2016*). The above results confirm the observations made by Ruohol et al. conducted in a group of 756 respondents (*Ruohola et al., 2006*). In a study evaluating the impact of dietary factors on BMD and the risk of fractures among female long-distance runners, it was shown that higher consumption of *e.g.* vitamin D supplementation, was accompanied by an increase in bone mineral density and fewer fractures (*Nieves et al., 2010*). Moreover, it was also emphasized that OC plays a significant role in energy metabolism, and its changes related to physical activity may potentially be an adaptation to the increased energy demand among athletes (*Banfi et al., 2010*). In the authors' study, the evaluation of diet did not show any significant influence on the research results.

In endurance sports competitions, physical performance depends, among others, on the size of energy resources. According to recommendations, the diet of people practicing endurance competitions (running) should provide 60–70% of energy from carbohydrates, 20–25% of energy from fats and 10–15% of energy from protein (*Jeukendrup, 2004*; *Jeukendrup, 2011*; *Nutrition and Athletic Performance, 2009*; *Kerksick et al., 2018*; *Vitale & Getzin, 2019*; *Malsagova et al., 2021*). The energy supply in the assessed food rations of the examined group of athletes was diversified. In some cases, it turned out to be insufficient to cover the 24-hour energy expenditure. In the authors' research, medium- and long-distance runners showed an increased share of fat in the diet of competitors (29.5 ± 3.0%), with a lower-than-recommended consumption of carbohydrates (53.9 ± 3.7%). Similar results were observed by *Tota et al. (2013)* who noted that the tested athletes had an insufficient supply of energy by an average of 400 kcal in relation to the demand, as well as an increased share of fat with lower consumption of carbohydrates in the diet of athletes. Within the context of exercise physiology, the significant role of carbohydrates in metabolic processes should be indicated (*Benardot, 2012*). In the discussed research, the differentiation was confirmed of some nutritional behaviours depending on the level of sport, expressed by practicing it or not, with an indication of more favourable choices in the group of runners training medium- and long-distance runs. The more rational eating behaviours of the training individuals was related to the choice of the right source of supplied energy. This behaviour is characteristic of people representing a higher sports level, which would indirectly indicate a higher level of their knowledge in the area of rational nutrition (*Kopeć et al., 2013*; *Gacek, 2017*).

The implementation of nutritional standards for selected vitamins and minerals is a frequent topic of research in various population groups due to their importance for the proper functioning of the body. The results of individual studies often indicate a deficiency of individual vitamins due to an inadequate diet or their excess (*Czapska et al., 2005*). In the authors' research, a low supply of some minerals and vitamins in the diet of active people was observed, which could indicate that people practicing medium- and long-distance running may be at risk of nutritional deficiencies. A similar tendency was demonstrated in the study by *Durkalec-Michalski, Baraniak & Jeszka (2015)* and *Tota et al.*

*(2013)*. The minerals found to be deficient are potassium and calcium. Probable causes of shortages include: in the elimination or small consumption of certain products, eating meals with too low nutritional density or insufficient amount of energy supplied with food (*Driskell & Wolinsky, 2005*; *Dunford & American Dietetic Association, 2006*). It seems that the exceeded supply of some minerals and vitamins along with the diet in nutritional practice is unavoidable. This is due to the recommended higher consumption of vegetables and fruits, wholegrain products and low-processed products.

Changes in the biomarkers described by us during the observation period allow us to hypothesize that the observation of the direction of changes in these indicators may prove important in monitoring bone-forming processes. Changes in the biomarkers described during the observation period allow to hypothesize that observing the direction of changes in these indicators may prove of significance in monitoring bone formation processes in men. A potential practical implication is the use of maximal efforts to activate bone remodelling processes. The selection of an optimal training load that regulates the mineralisation of bone tissue should contribute to preventing and reducing the risk of musculoskeletal injuries among people engaging in physical activity.

## CONCLUSIONS

A single-session physical effort performed at maximum intensity may become an effective tool to initiate positive changes in bone turnover markers after taking the type and intensity of exercise into account, as well as the age and sex of the subjects. In order to obtain consistent results in the future, factors directly affecting bone turnover should be closely monitored in order to understand the kinetic reactions taking place in bone tissue.

The authors are aware of some limitations resulting from this study. The long-term effect was not taken into account when determining markers of bone turnover, therefore, it cannot be clearly stated whether the anabolic effect was long-term or limited to the duration of exercise, as indicated in other reports (*Salvesen et al., 1994*; *Maïmoun et al., 2006*).

The sample size calculated for the study was relatively small and should be considered minimal. The presented results should be interpreted with caution and confirmed in a larger population.

### Funding
The project was financed under the programme of the Minister of Science and Higher Education under the name "Student Scientific Circles Create Innovations", project No. SKN/SP/498248/2021. The funders had no role in study design, data collection and analysis, decision to publish, or preparation of the manuscript.

### Grant Disclosures
The following grant information was disclosed by the authors:
Student Scientific Circles Create Innovations: SKN/SP/498248/2021.

## Competing Interests

The authors declare there are no competing interests.

## Author Contributions

- Małgorzata Bagińska conceived and designed the experiments, performed the experiments, analyzed the data, prepared figures and/or tables, and approved the final draft.
- Łukasz Marcin Tota conceived and designed the experiments, performed the experiments, analyzed the data, authored or reviewed drafts of the article, and approved the final draft.
- Małgorzata Morawska-Tota conceived and designed the experiments, performed the experiments, analyzed the data, prepared figures and/or tables, authored or reviewed drafts of the article, and approved the final draft.
- Justyna Kusmierczyk analyzed the data, prepared figures and/or tables, and approved the final draft.
- Tomasz Pałka analyzed the data, authored or reviewed drafts of the article, and approved the final draft.

## Human Ethics

The following information was supplied relating to ethical approvals (*i.e.*, approving body and any reference numbers):

The study project was approved by the Bioethics Committee at the District Medical Chamber in Kraków, No. 319/KBL/OIL/2021.

## Data Availability

The raw data is available in the Supplemental File.

## Supplemental Information

Supplemental information for this article can be found online at http://dx.doi.org/10.7717/peerj.17258#supplemental-information.

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
