# Peer review of "Changes in the concentration of bone turnover markers in men after maximum intensity exercise"

_PeerJ, doi:10.7717/peerj.17258_

## Round 0.1 · original submission · Major Revisions

Please make sure you clearly define your methods so that they are reproducible by others. In addition, make sure you clearly differentiate what is new about your paper and contextualize it with what has been done previously.

Reviewer 1 ·

Basic reporting

The introduction is succinct and coherent, yet it would benefit from providing additional context regarding the broader aspect of maximal intensity in exercise. To achieve this, incorporating relevant insights from running literature would be valuable. To enhance its depth, valuable insights from running literature, such as the study conducted by Kouvelioti et al. in 2018, could be incorporated, particularly regarding High-Intensity Interval Training (HIIT), which falls under the category of maximum intensity exercises. Further, the aim and purpose of this study needs to be expanded..line 46.

Experimental design

The Methods section is presented clearly, but it lacks sufficient detail regarding the participants, inclusion and exclusion criteria. Additionally, it is essential to elaborate on how the sample size was determined and provide specific measures for controlling the "control group." There is room for improvement in word choice, especially in lines 53 and 54.
Is the ELISA procedure performed in duplicate or triplicate? The authors need to provide information about the intra-assay and inter-assay coefficients of variation to assess its reliability.

Validity of the findings

The authors should disclose whether the observed difference is attributed to shifts in plasma volume changes, as this factor could significantly influence serum levels of bone markers. Moreover, in the discussion section, the authors should provide a more comprehensive explanation of OC and NTX responses, taking into account other relevant factors beyond just the duration. line 158
A graphical representation would be more effective than presenting the data in a table (Table 3). Need to explain more with other literature line 194. Statistical analysis should not be limited only within the Mann-Whitney U test.

Additional comments

Overall, this study comprised strong study design. However, the objectives the authors state in the beginning of their paper is not well supported by their findings.
• Differences in bone markers were examined between groups. However, there is no information on how the sample size were calculated, and power analysis were done. Effect sizes and %CV are not reported in the manuscript.
• This study also lacks providing information on research design.
• The authors provide comparable results to their findings from the literature. i.e., various studies which examined similar outcomes in various groups are described. However, the authors do not provide an explanation for the current study's results, specifically when they do not support previous findings. It is worth elaborating practical applications of this study.

Reviewer 2 ·

Basic reporting

No comment

Experimental design

In the introduction, the authors discuss general knowledge of bone turnover markers, but fail to specify the exercise intensity and duration used in previous findings. Moreover, they do not highlight the uniqueness and significance of their current study. It is essential to clearly outline how their research differs from previous work and its potential implications. The data collection procedures are not well-defined, particularly regarding bone turnover marker processes, such as intra/inter CV% (coefficient of variation), and whether study participants were fasted prior to the blood draw and after the exercise testing. These details are crucial for understanding the reliability and validity of the results.

Validity of the findings

No comment

Additional comments

No comment

---

## Round 0.2 · Minor Revisions

Reviewer 1 has offered additional suggestions. In addition, please carefully proofread to ensure the writing is consistent.

Reviewer 1 ·

Basic reporting

Line 93- 100 reword this sentence. Look for typos and grammatical errors.
Line 112 wording. Reword this sentence.
Line 122 Provide a reference for this paragaraph. Mention the effect size relevant in determining the sample size of the study.
Line 349 Explain and elaborate the practical application of this study.
Figure 2. Use pre and post rather than bef and after
There are lots of inconsistencies in the writing; please proofread.

Experimental design

Has addressed the previous comment significantly.

Validity of the findings

Well addressed; however, need to explain the practical application of this study.

---

## Round 0.3 · Minor Revisions

One of the original reviewers was finally able to provide some valuable feedback. The reviewer is highlighting the need to include some additional discussion about potential limitations of your design. You do not need to completely redo your paper, but consider implementing some of these suggestions for improvement. Thank you for your patience.

Reviewer 1 ·

Basic reporting

No Comment

Experimental design

No Comment

Validity of the findings

No Comment

Reviewer 2 ·

Basic reporting

1. Clarity and Depth of Literature Review:
The introduction provides a general overview of the importance of physical activity on bone health but lacks depth in summarizing previous findings on the acute effects of maximum-intensity exercise on bone turnover markers.
Suggested Improvement.
1) Suggested improvement: Expand the literature review to include a more detailed summary of existing research on the acute and long-term effects of different intensities of physical activity on bone turnover. This will help to contextualize the study's contributions and highlight its novelty.

Experimental design

1. Methodological Details
While the study design and methodology are generally well-described, certain aspects, such as the specific criteria for the control group's physical inactivity and the dietary assessment methods, are not sufficiently detailed.
1) Suggested Improvement: Provide a more comprehensive description of the control group selection criteria, ensuring it includes specific definitions of 'non-training' or 'sedentary' lifestyles. Additionally, elaborate on the dietary assessment methods used, including how dietary intake was quantified and how it was controlled or accounted for in the analysis.

2. Sample Size Justification
The manuscript mentions a power analysis was conducted, but details such as the assumed effect size and the rationale behind it are missing.
1) Suggested Improvement: Include a more detailed explanation of the power analysis, specifying the assumptions made (e.g., effect size, alpha level) and why these were deemed appropriate for this study. This will help readers assess the robustness of the study design.

3. Discussion of Findings
The discussion primarily focuses on the significance of the findings without adequately addressing potential limitations or the implications of these limitations on the study's conclusions.
1)Suggested Improvement: Enhance the discussion by critically evaluating the study's limitations, such as the short-term nature of the intervention, the small sample size, and potential dietary influences. Discuss how these factors might impact the generalizability and interpretation of the results and suggest directions for future research.

4. Nutritional Considerations
The manuscript briefly mentions that no effect of diet was noted on the changes in markers of bone turnover, but does not provide detailed data or analysis to support this statement.
1) Suggested Improvement: Provide a more detailed analysis of the dietary data collected, including how it was analyzed in relation to the bone turnover markers. Discuss whether the lack of observed dietary effects could be due to the study design, sample size, or other factors. If possible, include supplementary material with detailed dietary intake data and its correlation with the bone turnover markers.

Validity of the findings

No comment

Additional comments

The manuscript presents valuable research on the acute effects of maximum-intensity exercise on bone turnover markers in young men, filling a notable gap in the literature. However, there are several areas where improvements are needed to enhance the manuscript's clarity, depth, and scientific rigor. The literature review could be expanded to provide a more comprehensive context of existing research on physical activity's effects on bone turnover. Methodological details, particularly regarding the control group selection and dietary assessment methods, require further elaboration to ensure the study's robustness. A more detailed justification of the sample size and power analysis would strengthen the study design's credibility. In discussing the findings, a more critical examination of the study's limitations and their implications on the results is necessary. Additionally, incorporating a discussion on potential physiological mechanisms underlying the observed changes, as well as a deeper analysis of dietary data and its impact on bone turnover markers, would provide a more holistic understanding of the study's outcomes. Addressing these points will significantly improve the manuscript and contribute more effectively to the field of exercise science and bone metabolism.

---

## Round 0.4 · accepted · Accept

Comments have been addressed. The article is ready for publication.

Reviewer 2 ·

Basic reporting

The authors have adequately addressed my comments.

Experimental design

The authors have adequately addressed my comments.

Validity of the findings

The authors have adequately addressed my comments.

Additional comments

The authors have adequately addressed my comments.